# PRACTICAL INTEGRATION VIA SEPARABLE BIJECTIVE NETWORKS

**Christopher M. Bender[1,2], Patrick R. Emmanuel[2], Michael K. Reiter[3], Junier B. Oliva[1]**

[1]Department of Computer Science, The University of North Carolina
[2]The Johns Hopkins University Applied Physics Laboratory
[3]Department of Computer Science, Duke University
`{bender,joliva}@cs.unc.edu, patrick.emmanuel@jhuapl.edu,`
`michael.reiter@duke.edu`

## ABSTRACT

Neural networks have enabled learning over examples that contain thousands of dimensions. However, most of these models are limited to training and evaluating on a finite collection of *points* and do not consider the hypervolume in which the data resides. Any analysis of the model's local or global behavior is therefore limited to very expensive or imprecise estimators. We propose to formulate neural networks as a composition of a bijective (flow) network followed by a learnable, separable network. This construction allows for learning (or assessing) over full hypervolumes with precise estimators at tractable computational cost via integration over the *input space*. We develop the necessary machinery, propose several practical integrals to use during training, and demonstrate their utility.

## 1 INTRODUCTION

Most supervised learning problems operate by training a model on a finite collection, $\mathcal{T}$, of $N$ (typically paired) examples, $(x, y)$. The model is updated by comparing its predicted output to the expected output and performing some flavor of stochastic gradient descent based on the comparison and various regularizers. The process is repeated by reexamining the elements of $\mathcal{T}$ in a random order, possibly with augmentation, until the model parameters converge or an iteration budget is exceeded. This relatively simple procedure has proven to be remarkably effective in a variety of domains and these models have begun to permeate every aspect of modern science and everyday life (He et al., 2016; Silver et al., 2017; Brown et al., 2020).

The deep learning revolution has also resulted in highly effective generative models such as VAEs (Kingma & Welling, 2014), GANs (Goodfellow et al., 2014), and tractable likelihood models (Dinh et al., 2017; Oliva et al., 2018; Grathwohl et al., 2019). These models are largely used to create novel samples of impressive quality. In addition to sampling, likelihood models provide an estimate of the probability density function of the data which can be used for additional, downstream processes.

We *augment the training process* by constructing neural networks that allow for tractable integration over the *input domain*. This differs from implicit layers which utilize integration over a *parameterized* variable (Chen et al., 2018; Grathwohl et al., 2019). Access to fast and differentiable integrals allows us to regularize a model's *average* behavior using metrics that may not be available otherwise. Integration over the input space also allows us to supervise how the model behaves in *continuous regions that are not directly observed* in $\mathcal{T}$ and may even be out-of-distribution (OOD).

Alternative methods attempt to supervise examples outside of $\mathcal{T}$ by performing random perturbations (Gutmann & Hyvärinen, 2010), along the line between known examples (Zhang et al., 2018), or via a generative process (Zenati et al., 2018; Akcay et al., 2018). However, these methods are only capable of observing a small quantity of the total space. By integrating over entire regions, it is possible to observe a large portion of the space based on statistical relevance.

**Main Contributions**  We propose a general architecture that enables tractable integration over the input space, enabling supervision and custom regularization over continuous regions. We demonstrate how to construct this network and how it allows for a reduction in the computation cost required for dense numeric integration from exponential in the number of dimensions to linear. We derive several useful integrated formulations over continuous regions. Finally, we explore the impact of this architecture and regularizers on the standard accuracy and robustness to OOD examples on several standard classification datasets. The code utilized in this paper can be found at https://github.com/lupalab/sep_bij_nets.

**Notation**  Throughout this work, we consider the $M$ dimensional input features, $\mathbf{x} = [x_1, ..., x_M] \in \mathbb{R}^M$; the latent features, $\mathbf{z} \in \mathcal{Z} \subseteq \mathbb{R}^M$; and the $K$-wise classification probability, $y$. The input features $\mathbf{x}$ the training set, $\mathcal{T}_x$, are drawn from the in-distribution data, $\mathcal{D} \subseteq \mathbb{R}^M$. Subscripts represent a particular dimension, e.g., $\mathcal{D}_m$ corresponds to the $m^{\text{th}}$ dimension of the space. Paranthetical superscripts represent the subspace corresponding to a particular class, e.g., $\mathcal{D}^{(c)}$ is the subset of $\mathcal{D}$ where the data belongs to class $c$. The bijective network is given as $h$ such that $h : \mathcal{D} \to \mathcal{Z}$. Probability distributions over $\mathcal{D}$ and $\mathcal{Z}$ are given by $p$ with the corresponding subscript. Classification networks are given as $f$ and $g$. Variables with a "hat," $\hat{y}$, are predictions of the true quantity, $y$.

## 2 MOTIVATION

Neural networks are highly effective function approximators between two (typically) continuous spaces: $f : \mathcal{X} \to \mathcal{Y}$. However, networks are typically trained and evaluated using a finite collection of *points* without any explicit assessment of the complete hypervolume spanned by the data. This omission is understandable from an implementation perspective as the number of samples required to obtain a reasonable estimate over a volume scales exponentially with data dimensionality. However, human beings often have an understanding of how a process should behave *on average*. Ideally, we would like to embed this intuition into the model but currently cannot assess the average performance of a trained model outside of the held-out test set. Specifically, we would like to regularize the model by estimating the expected behavior of some metric, $\Omega$, produced by the model over the training data

$$\mathbb{E}_{\mathbf{x} \sim p(\mathbf{x})} \left[ \Omega(\hat{y}(\mathbf{x})) \right] = \int_{\mathcal{X}} \Omega(\hat{y}(\mathbf{x})) p(\mathbf{x}) d\mathbf{x}. \tag{1}$$

There are many useful choices of $\Omega$ over a variety of applications. If it is known what the model output should be on average ($\bar{y}$), we can construct $\Omega$ to encourage that behavior, e.g., $\Omega(\hat{y}) = (\bar{y} - \hat{y})^2$. Minimizing consistency metrics (Xie et al., 2020) are a common method to improve learning in label-starved problems. These encourage the model to produce similar outputs over neighborhoods around (labeled) examples from $\mathcal{T}_x$ where neighborhoods are created by random or domain-specific augmentations. This process can be viewed as an approximation to an integral over the neighborhood,

$$\mathbb{E}_{\epsilon \sim p(\epsilon)} \left[ \mathcal{L}(y, \hat{y}(\mathbf{x} + \epsilon)) \right] = \int \mathcal{L}(y, \hat{y}(\mathbf{x} + \epsilon)) p(\epsilon) d\epsilon \tag{2}$$

where $\mathcal{L}$ is a distance-like metric, and $\epsilon$ is the neighborhood. Equation 2 can be generalized to other neighborhoods. We can recast the standard classification problem as a discrete approximation to an integral. Typically, we minimize the cross-entropy between $\hat{y}(\mathbf{x}; \theta)$ and $y$ over the model parameters, $\theta$, for all $(\mathbf{x}, y) \in \mathcal{T}$ which becomes an integral over class-conditioned distributions, $p(\mathbf{x}|c)$,

$$\min_{\theta} - \sum_{x, y \in \mathcal{T}} \sum_k y_k \log (\hat{y}_k(x; \theta)) \Rightarrow \min_{\theta} - \sum_k \int_{\mathcal{D}^{(k)}} y_k \log (\hat{y}_k(x; \theta)) p(\mathbf{x}|k) d\mathbf{x}. \tag{3}$$

Unfortunately, integration in high dimension is difficult. Naive gridded solutions require an exponential number of points with error decreasing as $O(G^{-M})$, for $G$, $M$-dimensional points. Monte Carlo (MC) methods theoretically have better performance with error that decreases as $O(G^{-1/2})$. However, the rate of convergence for MC methods depends on the variance of samples (Veach, 1998), which may make for poor approximations in practice. Importance sampling (Bishop, 2006) can improve the performance of Monte Carlo methods to adapt to the regions with the greatest contribution.

We choose to model the data using a separable function. Separable functions have the key benefit of decomposing $M$-dimensional integrals into a combination of $M$ one-dimensional integrals. Each

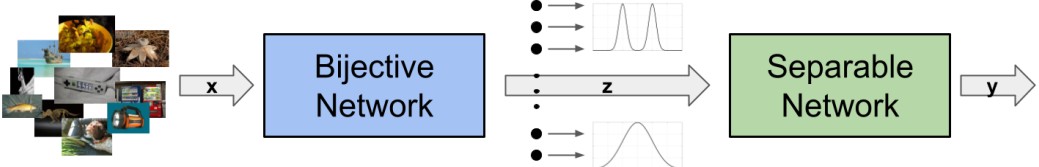

Figure 1: Depiction of the overall network with intervening distributions over the latent space, $\mathcal{Z}$

of these one-dimensional integrals can then be solved using any number of highly-accurate solvers (e.g., Runge-Kutta (Shampine, 2005), Dormand-Prince (Dormand & Prince, 1980), Tsit (Tsitouras, 2011), etc.) that have error rates better than $O(G^{-4})$ but are unavailable in high dimensions. The use of separable functions is a component of the VEGAS algorithm (Peter Lepage, 1978) and is utilized in conjunction with adaptive Monte Carlo sampling to approximate high-dimensional integrals.

The use of a separable function over the input space may make estimation of integrals over the model more accessible; however, they impose a strong, inappropriate inductive-bias. The obvious approach of utilizing a standard neural network as a feature extractor and integrating over learned features means that we would no longer have a valid integrator over the input space. We propose to solve this problem by utilizing bijective transforms prior to the separable network. The bijective transform allows us to decouple the data into a latent space where the data can be modeled using a separable network and guarantees equality between integrals in the latent space and integrals in the input space.

## 3 BACKGROUND

We perform integration over the input space by splitting neural network models down into two key components: (1) a *bijective* feature extractor, (2) a *separable* task network, see Fig. 1. For simplicity, we only consider classification tasks in this work. This makes our total network analogous with the common architecture where a classifier, often a linear layer or an MLP, is constructed on a feature extractor, such as a CNN. Unlike the typical process, we must place constraints on both networks so that we can integrate over the input domain. This network breakdown is similar to hybrid networks (Chen et al., 2019; Nalisnick et al., 2019b) except for the separability requirement on the classifier.

### 3.1 BIJECTIVE NETWORKS

Bijective networks are the key component in flow-based likelihood models. A bijective network, $h : \mathcal{D} \to \mathcal{Z}$, has a known forward and inverse operation so that data can exactly be reconstructed after the transformation. This allows for exact likelihood estimation via the change of variables formula:

$$\mathbf{z} = h(\mathbf{x};\ \theta), \quad \mathbf{x} = h^{-1}(\mathbf{z};\ \theta), \quad p_X(\mathbf{x}) = p_Z(h(\mathbf{x};\ \theta)) \left| \frac{\partial h}{\partial \mathbf{x}} \right| \tag{4}$$

where $p_Z$ is a predefined distribution over the latent space, often a standard Gaussian.

These models are trained via Eq. 4 to maximize the likelihood of the examples in $\mathcal{T}$. Once trained, flow-based likelihood models are commonly used as a generative process where samples are drawn from $p_Z$ and are then inverted through $h$ to arrive at an example in $\mathcal{D}$. Instead, we will take advantage of the fact that we can choose $p_Z$ and then utilize it in downstream tasks. Normalizing flow bijectors provide rich feature extractors that can represent a distribution of complicated inputs with simply-distributed, independent features, $z$. Given the independence and expressibility of these learned independent features, we build estimators using separable functions over $z$, which enables us to integrate over the data's domain while retaining expressibility.

The requirement for bijectivity places a strong constraint on network design, eliminating many common choices due to the need to maintain dimension or invert element-wise activations. Even naive convolutional operations become unavailable since they are not generally invertible. Modern advances have demonstrated methods to work around these limitations through the use of clever partitioning and coupling tricks Dinh et al. (2017) or the use of constraints Chen et al. (2019). However, the field of learnable bijective functions is less advanced then its unconstrained counterparts which results in reduced performance on auxiliary tasks. We utilize Glow Kingma & Dhariwal (2018) to process image data and continuous normalizing flows Grathwohl et al. (2019) for tabular data.

## 3.2 SEPARABLE FUNCTIONS

Separable functions have long been used in mathematics and physics to solve simple partial differential equations such as the homogeneous wave and diffusion equations (Strauss, 2007). We consider two types of separable functions, additive and multiplicative. All proofs can be found in Appendix A.

### 3.2.1 ADDITIVELY SEPARABLE FUNCTIONS

**Definition 3.1.** A function, $f : \mathbb{C}^M \to \mathbb{C}^K$, is additively separable if it is composed as a summation of element-wise functions operating independently on each dimension of the input:

$$f(\mathbf{v}) = \sum_{m=1}^{M} f_m(v_m; \phi_m) \tag{5}$$

**Theorem 3.1** (Additive Independent Integration). *Given an additively separable function, $f(\mathbf{v})$, an independent likelihood, $p(\mathbf{v}) = \prod_{m=1}^{M} p_m(v_m)$, and domain, $\mathcal{D}_v = \mathcal{D}_{v_1} \times ... \times \mathcal{D}_{v_M}$:*

$$\mathbb{E}_{\mathbf{v} \sim p(\mathbf{v})}[f(\mathbf{v})] = \sum_{m=1}^{M} \int_{\mathcal{D}_{v_m}} f_m(v_m) p_m(v_m) \, dv_m \tag{6}$$

### 3.2.2 MULTIPLICATIVELY SEPARABLE FUNCTIONS

**Definition 3.2.** A function, $g : \mathbb{C}^M \to \mathbb{C}^K$, is multiplicatively separable if it is composed as a product of element-wise functions operating independently on each dimension of the input:

$$g(\mathbf{v}) = \prod_{m=1}^{M} g_m(v_m; \psi_m) \tag{7}$$

**Theorem 3.2** (Multiplicative Independent Integration). *Given a multiplicitively separable function, $g(\mathbf{v})$, an independent likelihood, $p(\mathbf{v}) = \prod_{m=1}^{M} p_m(v_m)$, and domain, $\mathcal{D}_v = \mathcal{D}_{v_1} \times ... \times \mathcal{D}_{v_M}$:*

$$\mathbb{E}_{\mathbf{v} \sim p(\mathbf{v})}[g(\mathbf{v})] = \prod_{m=1}^{M} \int_{\mathcal{D}_{v_m}} g_m(v_m) p_m(v_m) \, dv_m \tag{8}$$

## 4 METHOD

Both forms of separable functions allow us to decompose a single $M$-dimensional integral into $M$ 1-dimensional integrals. A dense estimation of the integral without taking advantage of a separable function using $G$ points per dimension would require $\mathcal{O}(G^M)$ network evaluations. This is completely impractical for modern datasets where $M$ is at least on the order of hundreds and $G$ should be as large as possible. Exploiting separable functions allow us to reduce the complexity to $\mathcal{O}(GM)$.

However, it is unlikely that we could construct a separable function directly on the input space and achieve reasonable performance. Doing so would essentially require that each input dimension contribute to the final output independently of the others. Instead, we can combine Eq. 4 and Eq. 6 (alternatively, Eq. 8) to perform practical integration over the input domain while still allowing for inter-dependent contributions from each input dimension. To do so, we first let the latent distribution, $p_Z$, be independently (but not necessarily identically) distributed: $p_Z(z) = \prod_m p_m(z_m)$. This allows us to write the integral over the input space in terms of the latent space and then simplify via Eq. 6.

$$\int_{\mathcal{D}} f(h(\mathbf{x})) p_Z(h(\mathbf{x})) \left| \frac{\partial h}{\partial \mathbf{x}} \right| d\mathbf{x} = \mathbb{E}_{\mathbf{x} \sim p_X(\mathbf{x})}[f(h(\mathbf{x}))]$$

$$= \mathbb{E}_{\mathbf{z} \sim p_Z(\mathbf{z})}[f(\mathbf{z})] = \int_{\mathcal{Z}} p_Z(\mathbf{z}) f(\mathbf{z}) d\mathbf{z} = \sum_{m=1}^{M} \int_{\mathcal{Z}_m} f_m(z_m) p_m(z_m) \, dz_m \tag{9}$$

Each 1-dimensional integral can be easily approximated using a variety of integration approaches. In addition to the requirements placed on the feature extractor and task network, this formulation also

requires that we define the integration domain in the latent space as a Cartesian product of domains over each latent dimension. This may seem like a strong requirement since we apparently lose some level of interpretability; however, there are several advantages to defining the input domain in this learned space. Most notably, we know the data distribution in this space exactly and can tune the domain based on the goal of a particular integral.

Figure 2 contains a cartoon example that demonstrates how these methods combine to allow for integration over the complex data space. Both plots show the value of $f$; Fig. 2a shows the non-separable function in the input space and Fig. 2b shows the same data in the (separable) latent space, after the bijective transformation. The colored points and dotted lines are in correspondence between the two spaces and illustrate how the space is warped by the bijector to create a separable, independent latent space. The light gray lines represent the contours of the underlying probability distribution. We define both the integration domain and perform the integration in the latent space. For simplicity, we illustrate the integration using five, equally-spaced points (in both dimensions). The naive procedure would require sampling $f$ at each point in the grid (the blue points). The use of the separable functions allow us to integrate using only the red points and get the same estimate.

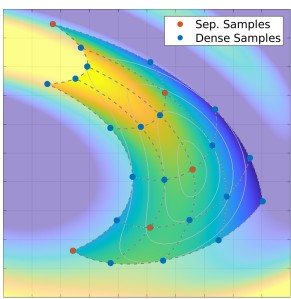

(a) Input Space $f(h(\mathbf{x}))$

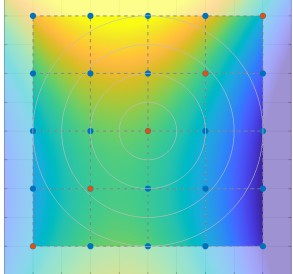

(b) Latent Space $f(\mathbf{z})$

Figure 2: Separable functions need $O(G)$ latent samples (red) instead of $O(G^M)$ input samples (blue). Integration regions emphasized.

## 5 PRACTICAL APPLICATIONS

In this section we discuss specific choices made when constructing the latent distribution, separable networks, and various integrals. For classification tasks, it is not possible to parameterize the normalized network output of class probabilities, $\hat{y}$, as a separable network since each prediction must sum to one. However, it is possible to parameterize each unnormalized logits as a separable network, e.g., $\hat{y}(\mathbf{x}) = \sigma\left(f(h(\mathbf{x}))\right)$, where $\sigma$ is the softmax operator and the output of $f$ are the unnormalized logits. While this limits what quantities can be integrated over exactly, it is still possible to integrate over approximations/bounds without resorting to brute-force methods.

**Out-of-Distribution Detection** We construct a global integration regularizer to provide resilience to out-of-distribution (OOD) examples. We enable OOD detection by introducing a "reject" option (Hendrickx et al., 2021) into the classification vector, increasing the number of classes by one, where the additional class indicates that the instance is OOD. An example is considered OOD if $\hat{y}_{K+1}$ exceeds a predefined threshold. We supervise the model over out-of-distribution examples by integrating the cross-entropy over the contrastive probability distribution, $q(\mathbf{z})$ (see Sec. 5.3.1).

**Semi-supervised Learning** We build a local integral to enhance consistency within latent neighborhoods and utilize it in place of pre-chosen augmentation strategies to inform a label-starved dataset. Specifically, we introduce a loss where all examples near a real, labeled example are regularized to share the real example's label as in Eq. 2 (see Sec. 5.3.2). We additionally use pseudo-labels Lee (2013) so that consistency is maintained about unlabelled points as the model grows more confident.

### 5.1 LATENT DISTRIBUTIONS

A critical component of this method is that the likelihoods over the latent space must be independent: $p_Z(\mathbf{z}) = \prod_m p_Z(z_m)$. As this is extremely unlikely to occur in the input space, we learn a bijective (flow) transformation from the input space to a latent space where the likelihood can be decomposed into independent components of our choice. Since we would like to use the latent features to discriminate between classes using a separable function, we choose to utilize a (bimodal) Gaussian mixture model. This allows the model to put

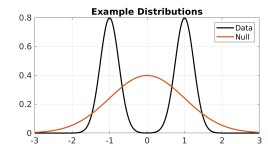

Figure 3: Data and contrastive distributions

different classes into one of two "buckets" per feature and enables classification through multiple features. This results in an exponential number of components (with respect to dimension) in the full latent space, e.g., $2^M$ components. However, we can easily offset this by choosing some dimensions to use a unimodal latent distribution while the remaining dimensions are still bimodal:

$$p_{Z_m}(z_m) = \begin{cases} 0.5\,\mathcal{N}(-\mu, \sigma_1^2) + 0.5\,\mathcal{N}(\mu, \sigma_1^2), & \text{if } m \leq L \\ \mathcal{N}(0, \sigma_2^2), & \text{else} \end{cases} \tag{10}$$

In addition to the typical latent data distribution, $p_Z(\mathbf{z})$, we explicitly include a contrastive distribution, $q(\mathbf{z})$. This distribution is critical if we desire to regularize our model outside the data distribution, i.e., setting some background/default behavior. When using a bimodal data distribution, we additionally desire that latent vectors are more likely under this distribution once the vector is sufficiently far away from any single in-distribution mode. Therefore, we select this distribution so that it fully encapsulates all the components of the data distribution, e.g., $q_m(z_m) > p_{Z_m}(z_m)$ for $||z_m| - \mu| > \gamma$. When the data distribution is unimodal we choose $p_m = q_m$ so that the feature is uninformative.

We utilize a standard Gaussian for the contrastive distribution and set $\mu$ to 1 and experiment with $\sigma_1 \leq 0.5$. This choice allows us to formulate OOD examples that exist between the in-distribution data as near zero, and outside the in-distribution data, near the tails. Figure 3 illustrates the data and contrastive distributions for a latent feature for convenience.

## 5.2 SEPARABLE NETWORKS

We explore three different formulations of separable networks. The first formulation learns a distinct quadratic function, the second learns a symmetric hinge, and the third learns a multi-layer perceptron. These functions have distinct parameterizations for each latent feature and each in-distribution class. The out-of-distribution logit is held fixed at zero: $l_{K+1}(\mathbf{x}) = 0$.

$$f_{k,m}^{(\text{quad})}(z_m; \alpha_{k,m}, u_{k,m}, \nu_{k,m}) = \alpha_{k,m} - \frac{(z_m - u_{k,m})^2}{2e^{2\nu_{k,m}}} \tag{11}$$

$$f_{k,m}^{(\text{hinge})}(z_m; \alpha_{k,m}, u_{k,m}, \nu_{k,m}) = \alpha_{k,m} - |z_m - u_{k,m}|\,e^{\nu_{k,m}} \tag{12}$$

where $k$ is the logit class index and $m$ is the feature dimension so that the total unnormalized logit, $l_k(\mathbf{z})$, is $l_k(\mathbf{z}) = \sum_m f_{k,m}(z_m)$. The separable MLP is constructed by concatenating a learned vector per feature and class to each 1-dimensional feature and constructing a standard MLP that operates on that augmented state. This formulation provides the greatest flexibility as it leverages all the benefits of the universal approximator theorem (Hornik, 1991). Unfortunately, we find the MLP version difficult to train in practice, often resulting in degenerate solutions. Fixing the OOD logit at zero provides a fixed reference point for the other logits. We interpret this as each latent feature voting on how in-distribution the example is on a per-class basis.

## 5.3 INTEGRALS

The cross-entropy, $\text{CE}(\hat{y}(\mathbf{x}), y)$, between the normalized model output, $\hat{y}(\mathbf{x})$, and the true labels, $y$, is commonly used to train classification models. In this section, we explore global and local integrals of the cross-entropy loss used to train most classifiers (see Eq. 3) when $\hat{y}$ is given as the composition of the softmax function, a separable function and a bijective function, e.g., $\sigma(f(h(\mathbf{x})))$. We can express this as $\min_\theta \sum_{\mathbf{x}, y \in \mathcal{T}} \text{CE}(\hat{y}(\mathbf{x}), y)$ owing to the finite number of elements in $\mathcal{T}$. Ideally, we would minimize the model cross-entropy over all possible examples in $\mathcal{D}$. We accomplish this by defining the subspace, $\mathcal{D}^{(c)} \subset \mathcal{D}$, that has a given label and corresponding conditional distribution, $p(\mathbf{x}|c)$

$$\mathbb{E}_{\mathbf{x} \sim p(\mathbf{x}|c)}\left[\text{CE}\left(\hat{y}(\mathbf{x}), \mathbf{c}\right)\right] = -\mathbb{E}_{\mathbf{x} \sim p(\mathbf{x}|c)}\left[\log(\hat{y}_c(\mathbf{x}))\right] = -\int_{\mathcal{D}^{(c)}} \log(\hat{y}_c(\mathbf{x}))\,p(\mathbf{x}|c)d\mathbf{x}. \tag{13}$$

It is not possible to represent $\hat{y}(\mathbf{x})$ as separable network due to the softmax operation but it is possible to parameterize the unnormalized logits as a separable network. Substituting this parameterization into Eq. 13 and transforming to the latent space yields a bound for the expected cross-entropy

$$\mathbb{E}_{\mathbf{x} \sim p(\mathbf{x}|c)}\left[\text{CE}\left(\hat{y}(\mathbf{x}), y\right)\right] \leq -\sum_{m=1}^{M} \int_{\mathcal{Z}^{(c)}} f_{c,m}(z_m)p_m(z_m|c)dz_m$$
$$+ \log\left(\sum_{j=1}^{K+1} \prod_{n=1}^{M} \int_{\mathcal{Z}^{(c)}} \exp\left(f_{j,n}(z_n)\right)p_n(z_n|c)dz_n\right). \tag{14}$$

See Appendix E for a full derivation. We will utilize this formulation in conjunction with a contrastive prior to supervise OOD examples, Sec. 5.3.1, and with a local prior to enforce consistency, Sec. 5.3.2.

### 5.3.1 OUT-OF-DISTRIBUTION SUPERVISION

We can utilize the cross-entropy integral in different ways by choosing the label we would like to apply over a domain. For example, we can integrate the cross-entropy over the OOD latent space, $\mathcal{U}$, with the true label fixed to the reject class ($c{=}K{+}1$), and the latent distribution over codes is the contrastive distribution (Sec. 5.1), $p(z|c{=}K{+}1) = q(z)$; using Eq. 14 with $f_{K+1,n}{=}0$:

$$\mathcal{L}_{\text{GLBL}} = \log\left(\sum_{j=1}^{K}\prod_{n=1}^{M}\int_{\mathcal{U}}\exp\left(f_{j,n}(z_n)\right)q_n(z_n)dz_n\right). \tag{15}$$

In effect, Eq. 15 discourages OOD data from being labeled as any in-distribution label. Since the data and contrastive distributions overlap, the model could degenerate and always make OOD decisions; however, this would be in conflict with the standard cross-entropy loss applied to each example in the training set. So long as $\mathcal{L}_{\text{GLBL}}$ is not weighted too strongly, the model achieves equilibrium by making OOD predictions over regions where the contrastive distribution is more likely. In effect, we supervise OOD training without requiring any OOD data.

### 5.3.2 LOCAL CONSISTENCY

We may also perform integration over neighborhoods of data points in $\mathcal{T}_x$ and utilize that point's label over the entire region. This integral serves to improve consistency around observations. Adversarial attacks (Goodfellow et al., 2015; Madry et al., 2018) are often constructed as perturbations on real data points and adversarial training finds one such point and includes it in the training set. We can interpret local consistency integrations as a form of *average* adversarial training. We reformulate the integral to be centered around a particular datum, $\mathbf{x}_0$ and its class label, $y$

$$\mathcal{L}_{\text{LCL}} = -\sum_k y_k \int_{\mathcal{V}}\log(\sigma\left(f_k(h(\mathbf{x}_0) + \mathbf{v})\right))p_V(\mathbf{v})d\mathbf{v} \tag{16}$$

A complication resulting from local integration is how to select the local distribution, $p_V(\mathbf{v})$, and the neighborhood, $\mathcal{V}$. There are many reasonable choices depending on the goal of the local integration. For simplicity, we choose each $\mathcal{V}_m \in [-\varepsilon, \varepsilon]$ and $p_m(v_m)$ to be uniformly distributed.

### 5.4 LOSS COMPONENTS

The final training loss used to evaluate the model is composed of different combinations of the standard cross-entropy loss over class predictions, $\mathcal{L}_{\text{CE}} = -\sum_k y_k \log(\hat{y})$, the negative log-likelihood loss over in-distribution data points, $\mathcal{L}_{\text{NLL}} = -\log(p_{\mathcal{Z}}(h(\mathbf{x}))) - \log\left|\frac{\partial h}{\partial \mathbf{x}}\right|$, and the integration losses, $\mathcal{L}_{\text{GLBL}}$ and $\mathcal{L}_{\text{LCL}}$. We always include $\mathcal{L}_{\text{CE}}$ and $\mathcal{L}_{\text{NLL}}$. Based on results from previous hybrid networks (Chen et al., 2019; Nalisnick et al., 2019b), we weight the negative log-likelihood by $1/M$, which is analogous to using bits per dimension and introduce an additional weight over the cross-entropy, $\lambda$, which controls the relative importance of the generative and classification tasks. When included, each integration penalty shares the same weight as $\mathcal{L}_{\text{CE}}$ with additional binary weight, $\pi$

$$\mathcal{L}_{\text{total}} = \frac{1}{M}\mathcal{L}_{\text{NLL}} + \lambda\left(\mathcal{L}_{\text{CE}} + \pi_{\text{GLBL}}\mathcal{L}_{\text{GLBL}} + \pi_{\text{LCL}}\mathcal{L}_{\text{LCL}}\right). \tag{17}$$

## 6 RELATED WORK

**Noise Contrastive Distributions** Noise contrastive methods introduce a distribution that is distinct from the data distribution. The constrastive distribution can be learned and provides a mechanism to supervise the model to discriminate between true and contrastive samples (Gutmann & Hyvärinen, 2010). This forces the model to learn discriminative statistical properties of the data. We utilize this idea to inform the null-space over which we will integrate. We fix the data and contrastive distributions and learn the bijective map between input and latent spaces.

**Hybrid Models**   Normalizing flows are a family of flexible, tractable likelihood estimators based on the application of learnable, bijective functions Dinh et al. (2017); Grathwohl et al. (2019). These methods have shown strong performance, both as likelihood estimators and as generative processes. Recent work has coupled advances in bijective networks to construct invertible feature extractors that are capped by a more conventional classifier. The combination bijective/classifier structure are called *hybrid models* and show reasonable performance as likelihood and discriminative models Chen et al. (2019); Nalisnick et al. (2019b). Unfortunately, the discriminative performance of these models is lower than traditional methods. We utilize hybrid models with constraints that enable tractable integration. Other hybrid models architectures are less restricted but prohibits practical integration.

**Out of Distribution Detection**   OOD detection is a challenge for many modern ML algorithms. Recent work has demonstrated that OOD data can achieve higher likelihoods than the training data Hendrycks et al. (2019); Nalisnick et al. (2019a). Several recent methods have been developed to detect OOD examples including Maximum Softmax Probability (MSP) Hendrycks & Gimpel (2017), Outlier Exposure (OE) Hendrycks et al. (2019), Multiscale Score Matching (MSMA) Mahmood et al. (2021), ODIN Liang et al. (2018), and Certified Certain Uncertainty (CCU) Meinke & Hein (2020). Lee et al. (2018) utilize a GAN trained with a classifier to produce examples near but outside the training distribution. These methods are constructed specifically for OOD detection whereas our method is applicable to a variety of problems.

**Semi-supervised Learning**   Semi-supervised learning is a burgeoning research area that learns from a large data corpus when only a small subset are labeled. The problem setting is very pertinent as it is often easy to acquire data examples but can be extremely time consuming to create the corresponding labels. Many modern methods can achieve very strong performance with very few labels Zhang et al. (2018); Chen et al. (2020). However, most of these methods rely on domain-specific augmentation strategies that are difficult to replicate in new data regimes, i.e., for non-image data. Fortunately, methods such as SSVAE Kingma et al. (2014) and VIME Yoon et al. (2020) are domain agnostic. These methods are built exclusively for semi-supervised learning but we only require an additional regularizing penalty to achieve comparable performance on tabular datasets.

## 7 EXPERIMENTS

All models were constructed using PyTorch (Paszke et al., 2019), trained using PyTorch-Lightning (Falcon, 2019), utilized bijectors and distributions from Pyro (Bingham et al., 2018), and were trained using Adam (Kingma & Ba, 2015). We assess the integrable model's performance in a semi-supervised regime and against OOD examples. See Appendix B and C for additional experiments and training details.

### 7.1 SPIRALS

We construct a synthetic, 2D dataset composed of three intertwined spirals, see Fig. 4a. Suppose that, due to an unknown sampling bias, we only observe two of the three arms (missing the green arm) and constructed our model as a two-class problem with a third, reject class.

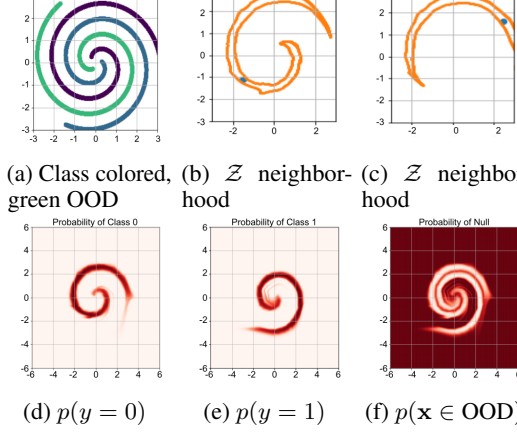

(a) Class colored, green OOD

(b) $\mathcal{Z}$ neighborhood

(c) $\mathcal{Z}$ neighborhood

(d) $p(y = 0)$

(e) $p(y = 1)$

(f) $p(\mathbf{x} \in \text{OOD})$

Figure 4: Three-arm spirals results

We sample points in a dense 2-dimensional grid at twice the range of the data in both dimensions and evaluate the probability that each point belongs to the two known classes and the OOD class. Figures 4d and 4e illustrate the probability of belonging to the two known classes and Fig. 4f contains the probability that each point is OOD. Red and white values indicate high and low probabilities, respectively. The model successfully identifies the two in-distribution regions. Unsurprisingly, the model also identifies data outside of the training data range as being OOD and, impressively, identifies the unobserved spiral and the region between spirals as being OOD.

Finally, we sample a square box (orange) around a data point (blue) in the latent space and invert the box to assess the appropriateness of the neighborhood in the input space and plot the result, Figs. 4b and 4c. As expected, the bijector converts the latent Euclidean neighborhood into a semantically meaningful neighborhood. Had we included the local cross-entropy integral in this training process, all points within the orange boundary would have received the same label as the data point.

## 7.2 OUT OF DISTRIBUTION DETECTION

We test how our models respond to OOD examples when trained with global integrals over the noise-contrastive prior and consistency integrals and compare to methods designed cfor OOD detection. See Appendix B.3 for standard performance and Sec. 6 for a discussion of the baselines. Table 1 contains the AUPR for the various methods against similar but different datasets (see Appendix B.4 for the AUROC). We juxtapose SVHN vs CIFAR10 and MNIST vs FMNIST plus Extended MNIST (EMNIST) (Cohen et al., 2017). We include a separable, hybrid model without integral regularizations (hybrid) and the same model with regularizations (Int. Reg.) to illustrate the utility of learning over hypervolumes. When MNIST or FMNIST are the in-distribution set, the regularized, integrable network performs on par with the best baseline methods. SVHN and CIFAR10 show reasonable OOD performance but are not as strong as the baselines. This is not surprising since the integrals rely on a reasonable estimate of $p(\mathbf{x})$, which both datasets have failed to achieve, Table 5.

Table 1: Area under the PR curve (percentage).

| In | Out | GAN | ODIN | MSMA | OE | CCU | Hybrid | Int. Reg. |
|---|---|---|---|---|---|---|---|---|
| MNIST | FMNIST | 99.4 | 98.8 | - | 99.9 | 99.9 | $93.7 \pm 6.8$ | $99.7 \pm 0.17$ |
| | EMNIST | 84.5 | 78.4 | - | 91.4 | 84.3 | $97.2 \pm 1.9$ | $99.8 \pm 0.03$ |
| FMNIST | MNIST | 99.9 | 99.2 | 80.8 | 97.0 | 98.3 | $63.9 \pm 8.3$ | $95.4 \pm 0.04$ |
| | EMNIST | 100. | 99.3 | - | 98.6 | 99.1 | $93.8 \pm 3.5$ | $98.8 \pm 0.47$ |
| SVHN | CIFAR10 | 98.6 | 97.3 | 92.5 | 100. | 100. | $71.8 \pm 2.0$ | $76.8 \pm 1.4$ |
| CIFAR10 | SVHN | 80.5 | 92.7 | 99.0 | 98.5 | 97.5 | $84.6 \pm 0.8$ | $87.0 \pm 2.1$ |

## 7.3 SEMI-SUPERVISED LEARNING

We apply the local integral to the separable hybrid model and train on several tabular datasets with 10% of the labels and domain-specific augmentation strategies are unavailable. These models do not utilize the OOD class or global penalty. We apply pseudo-labelling with thresholds of 0.9-0.95. Table 2 compares the perfor-

Table 2: Tabular dataset semi-supervised accuracy (%).

| | SSVAE | VIME | Local Int. |
|---|---|---|---|
| Flat MNIST | 88.9 | 95.8 | $94.9 \pm 0.16$ |
| MiniBooNE | 92.2 | 91.7 | $93.5 \pm 0.074$ |
| HepMass | 83.1 | 82.0 | $85.4 \pm 1.2$ |

mance of the integrated hybrid models to several standard (non-image) semi-supervised baselines on flat MNIST (MNIST, flattened to a vector), MiniBooNE Bazarko (2001), and HepMass Baldi et al. (2016). We utilize a CNF Grathwohl et al. (2019) as the bijector and the hinge as the classifier. We see that the integrated model achieves similar-to-better performance than the other methods on all datasets, showcasing the generality of integrated regularizers in different problem spaces.

## 8 CONCLUSIONS

In this work, we develop architectures that enable tractable integration over the data space. We demonstrate that the ability to supervise regions and not isolated points encourages the model to learn better representations and be less likely to degenerate. We consider several formulations that allow us to regularize the model's behavior based on consistency and contrast without relying on augmentations of and sparse comparisons between a finite collection of data points. We experiment with the various integrals to obtain promising out-of-distribution detection. Through now tractable integrals, this work enables future methods and applications for learning over continuous regions.

## ACKNOWLEDGMENTS

This research was partly funded by grants NSF IIS2133595 and NSF 2113345 and by NIH 1R01AA02687901A1.

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

## A  PROOFS

### A.1  ADDITIVELY SEPARABLE FUNCTIONS

**Theorem A.1** (Additive Independent Integration). *Given an additively separable function, $f(\mathbf{v})$, an independent likelihood, $p(\mathbf{v}) = \prod_{m=1}^{M} p_m(v_m)$, and domain, $\mathcal{D}_v = \mathcal{D}_{v_1} \times ... \times \mathcal{D}_{v_M}$:*

$$\mathbb{E}_{\mathbf{v} \sim p(\mathbf{v})}\left[f\left(\mathbf{v}\right)\right] = \sum_{m=1}^{M} \int_{\mathcal{D}_{v_m}} f_m\left(v_m\right) p_m\left(v_m\right) dv_m \tag{A1}$$

*Proof.*

$$\int f_n\left(v_n\right) p_m\left(v_m\right) dv_m = \begin{cases} f_n\left(v_n\right), & \text{if } n \neq m \\ \int f_n\left(v_n\right) p_n\left(v_n\right) dv_n, & \text{else} \end{cases} \tag{A2}$$

$$\mathbb{E}_{\mathbf{v} \sim p(\mathbf{v})}\left[f\left(\mathbf{v}\right)\right] = \int_{\mathcal{D}} \left(\sum_{n=1}^{M} f_n\left(v_n\right)\right) \left(\prod_{m=1}^{M} p_m(v_m)\right) d\mathbf{v} \tag{A3}$$

$$= \sum_n \int_{\mathcal{D}} f_n\left(v_n\right) \prod_{m=1}^{M} p_m(v_m) \, d\mathbf{v} \tag{A4}$$

Substituting Eq. A1 into Eq. A3 reduces the $M$-dimensional integral over independent likelihoods into a 1-dimensional integral and completes the proof. □

### A.2  MULTIPLICATIVELY SEPARABLE FUNCTIONS

**Theorem A.2** (Multiplicative Independent Integration). *Given a multiplicitively separable function, $g(\mathbf{v})$, an independent likelihood, $p(\mathbf{v}) = \prod_{m=1}^{M} p_m(v_m)$, and domain, $\mathcal{D}_v = \mathcal{D}_{v_1} \times ... \times \mathcal{D}_{v_M}$:*

$$\mathbb{E}_{\mathbf{v} \sim p(\mathbf{v})}\left[g\left(\mathbf{v}\right)\right] = \prod_{m=1}^{M} \int_{\mathcal{D}_{v_m}} g_m\left(v_m\right) p_m\left(v_m\right) dv_m \tag{A5}$$

*Proof.* Since each component of $g_n$ is matched to a component in $p_m$, the two products can be recombined to isolate like-dimensions. Each dimension can then be integrated independently.

$$\mathbb{E}_{\mathbf{v} \sim p(\mathbf{v})}\left[g\left(\mathbf{v}\right)\right] = \int d\mathbf{v} \left(\prod_{n=1}^{M} g_n(v_n)\right) \left(\prod_{m=1}^{M} p_m(v_m)\right)$$

$$= \int \cdots \int \left(\prod_{n=1}^{M} g_n(v_n) p_n(v_n)\right) dv_1 \cdots dv_M$$

$$= \int g_1(v_1) p_1(v_1) dv_1 \cdots \int g_M(v_M) p_M(v_M) dv_M$$

□

## B  ADDITIONAL EXPERIMENTS

All image datasets were trained using Glow as the bijector with the number of levels adjusted for the size of the image. MNIST and Fashion MNIST were trained with Adam for 50 epochs with an exponentially decaying learning rate of $\gamma = 0.75$. SVHN and CIFAR10 utilized a similar training procedure but with 75 epochs and a slower exponential decay of $\gamma = 0.99$. We find that utilizing an average instead of a sum over separable features and soft-thresholding at $-1$ improves training stability.

## B.1 ADVERSARIAL ROBUSTNESS

We test the efficacy of the local cross-entropy integral by utilizing a synthetic dataset from (Tsipras et al., 2019). The data is constructed via

$$y \sim \{-1,\ +1\}, \qquad x_1 = \begin{cases} +y, & \text{w.p. } p \\ -y, & \text{w.p. } 1-p \end{cases}, \qquad x_2, ..., x_{M+1} \sim \mathcal{N}(\eta y, 1) \qquad \text{(B1)}$$

and the adversarial examples are constructed analytically by, essentially, flipping the sign of $\eta$. We utilize this synthetic dataset to test adversarial robustness because it provides an upper bound on the adversarial accuracy relative to the standard accuracy which allows us to quantify our performance with respect to an absolute. Additionally, because adversarial examples can be constructed exactly without relying on an optimization procedures, there can be no concerns that this defense relies on obfuscated gradients (Athalye et al., 2018). We refer the interested reader to the original paper for a thorough discussion.

We choose $M = 10$, $p = 0.95$, and $\eta = 2/\sqrt{M}$. For these choices, a strong classifier will obtain near perfect standard accuracy with zero adversarial accuracy while a robust classifier will obtain 95% standard and adversarial accuracy. We use a similar architecture to that found in Sec. C.1 except we utilize ten blocks instead of five. We train one model with only the standard cross-entropy and negative log-likelihood losses and a second model with the additional global and local cross-entropy integration loss. We find that the model with only the standard losses achieves reasonably good standard performance and fairly poor adversarial

Table 3: Std. & Adv. Accuracy

|            | Std.  | Adv. |
|------------|-------|------|
| Accurate   | 100.0 | 0.0  |
| Robust     | 95.0  | 95.0 |
| Baseline   | 97.9  | 27.5 |
| Integrated | 94.5  | 90.8 |

accuracy. The model with the additional integration losses contains considerably better adversarial performance, nearing the upper and, necessarily lower, standard accuracy bound. Table 3 summarizes our results and the ideal performances.

## B.2 TOY SEMI-SUPERVISED REGRESSION

To illustrate the utility of global regularizations, we construct a one-dimensional, semi-supervised problem with $x \sim \mathcal{N}(0, 4)$ and $y = \tanh(x)$. We keep all values of $x$ but remove $y$ values corresponding to negative $x$ during training. Unlike the standard semi-

Table 4: Semi-supervised integration regularization

| $\mathbb{E}[y] = 0$ | Standard | Integrated |
|---------------------|----------|------------|
| Sup. MSE   | 1.195e-5 $\pm$ 1.882e-5 | 2.487e-5 $\pm$ 1.769e-5 |
| Unsup. MSE | 1.399 $\pm$ 1.198 | 0.06187 $\pm$ 0.04538 |
| $\mathbb{E}[\hat{y}]$ | 0.1041 $\pm$ 0.08582 | 8.570e-4 $\pm$ 4.150e-4 |

supervised problem, the missing labels are not random but are the result of limitations or bias in the data collection process. We train two models that are identical except that the integrated model includes a penalty over $\mathbb{E}[\Omega(\hat{y}(\mathbf{x}))]$ based on foreknowledge that the average value is zero. Table 4 shows how the models perform over the supervised and unsupervised regions. The inclusion of the integration regularizer allows the model to reason about regions that it has not directly observed and has decreased the error in that region by two orders of magnitude.

### B.3 STANDARD PERFORMANCE

We test the impact of integrable models on OOD performance against several standard image classification datasets: MNIST (Deng, 2012), Fashion MNIST (FMNIST) (Xiao et al., 2017), SVHN (Netzer et al., 2011), and CIFAR10 (Krizhevsky et al., 2009). See Appendix B for architures, distributions, and training parameters. Table 5 contains the validation standard accuracy and bits per dimension for all datasets with all integration regularizers. The upper portion of the table is averaged over 10 runs and contains a reject option in the separable model. The lower portion is a single run without a reject option or any integrated regularizers. We find that the model performs reasonably well for MNIST and

Table 5: Standard accuracy and bits-per-dimension for different datasets

|        | Acc.          | BPD             |
|--------|---------------|-----------------|
| MNIST  | $98.3 \pm 0.2$ | $2.54 \pm 0.13$ |
| FMNIST | $88.1 \pm 2.3$ | $4.76 \pm 0.64$ |
| CIFAR10 | $73.4 \pm 1.8$ | $5.62 \pm 0.76$ |
| SVHN   | $89.9 \pm 0.7$ | $4.14 \pm 0.08$ |
| CIFAR10* | 76.5        | 3.78            |
| SVHN*  | 90.0          | 2.24            |

FMNIST but the integrated losses cause a large degradation in performance for SVHN and CIFAR10. The removal of these losses produces similar accuracies but much-improved BPD, consistent with the hybrid network results reported by ResidualFlows (Chen et al., 2019) when Glow is used.

### B.4 OUT OF DISTRIBUTION DETECTION AUROC COMPARISONS

Table 6: Area under the ROC curve (percentage)

|         |         | GAN  | ODIN | MSMA | OE   | CCU  | Hybrid          | Int. Reg.        |
|---------|---------|------|------|------|------|------|-----------------|------------------|
| MNIST   | FMNIST  | 99.4 | 98.7 | -    | 99.9 | 99.9 | $93.6 \pm 6.5$  | $99.8 \pm 0.12$  |
|         | EMNIST  | 92.8 | 88.9 | -    | 95.8 | 92.0 | $77.7 \pm 12.6$ | $98.2 \pm 0.28$  |
| FMNIST  | MNIST   | 99.9 | 99.0 | 82.6 | 96.3 | 97.8 | $69.8 \pm 9.2$  | $95.4 \pm 3.1$   |
|         | EMNIST  | 99.9 | 99.3 | -    | 99.3 | 99.5 | $63.4 \pm 18.2$ | $87.8 \pm 4.4$   |
| SVHN    | CIFAR10 | 96.8 | 95.9 | 97.6 | 100. | 100. | $87.0 \pm 0.7$  | $91.3 \pm 0.6$   |
| CIFAR10 | SVHN    | 83.9 | 96.7 | 99.1 | 98.8 | 98.2 | $73.5 \pm 1.8$  | $78.85 \pm 2.7$  |

### B.5 INTERPRETABILITY

The separably model and independent latent dimensions allows us to reason about how the model is making decisions in the latent space. Unfortunately, for most bijectors, it is not possible to carry this interpretability back to the input space. Figure 5 demonstrates the final state of the model trained on Fashion MNIST for several features with respect to the latent distribution, per in-distribution class and juxtaposed with the out of distribution data. Specifically, the top row contains the logit components, $f_{k,m}$, learned by the separable network, color-coded by class; the middle row contains the distribution of each class (colors matched to logits); and the bottom row contains the distribution of the in-distribution data (blue) and OOD data (red). The top row illustrates how the classification network makes its decisions per feature over the distribution presented in the middle row. We see that the logit components map well to the distribution of the data in each feature and provides some intuition for how certain the classifier is over a feature. This demonstrates how this architecture allows for reasonable interpretibility from the latent space. Any value above zero (the dotted black line) is considered in-distribution and the most likely class is the line with the greatest value at that point. The features were chosen to demonstrate the diversity of the learned solution over features. Generally, the data maps reasonably well to the bimodal distribution, though we do occassionally see mode collapse as in Fig. 5e. Fortunately, in these cases the logits tend to be fairly uninformative and only introduce a small bias. Figures 5a through 5c show a common trend where the OOD data has heavy overlap with one of the two clusters but not the other. While we do see some diversity amongst the in-distribution classes that overlap with the OOD data the "bags" (gray) and "sandals" (cyan) class overlap most often. Finally, Fig. 5d demonstrates a latent feature where the OOD data falls in the region between the two data components.

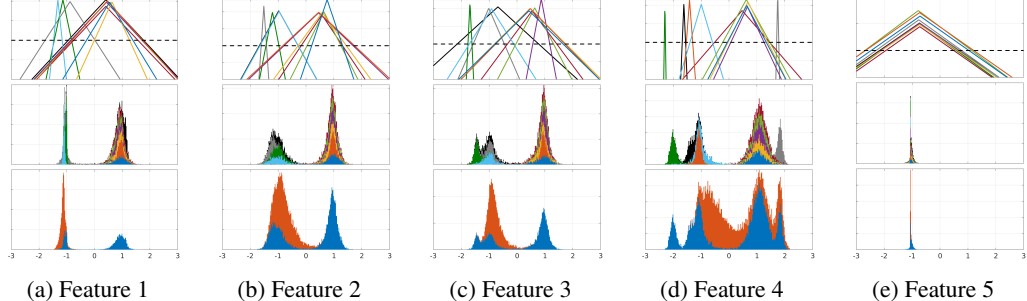

| (a) Feature 1 | (b) Feature 2 | (c) Feature 3 | (d) Feature 4 | (e) Feature 5 |

Figure 5: Each subplot corresponds to a single latent feature. The top and middle rows contain the unnormalized logit components and distributions per class, with matching colors. The bottom row compares the distribution of Fashion MNIST (blue) and MNIST (red). x-axis shared across all rows.

### B.6 MNIST LEAVE-ONE-OUT

Following (Ahmed & Courville, 2020), we test the robustness of our method against semantic OOD examples by training models on all but one class from MNIST (Deng, 2012) and assessing OOD performance on the held out class. We repeat the process for each class and report the AUROC and AUPR. In all cases, we train using ten different random seeds and report the average performance and standard deviation across seeds that do not degenerate against the in-distribution validation set. Table 8 and Table 7 contain the results of our method using both integrated losses compared to several relevant baselines. Results from Efficient GAN-Based Anomaly Detection (EGBAD) (Zenati et al., 2018) and GANomaly (Akcay et al., 2018) are taken by estimating the results from figures in both works. The Semantic Anomalies (Ahmed & Courville, 2020) results are obtained by executing the official version of the code using the rotation auxiliary task with two different batch sizes (128 and 256). The Semantic Anomalies results in both tables are the best across both batch sizes and detection methods (ODIN (Liang et al., 2018) and MSP (Hendrycks & Gimpel, 2017)) based on the AUPR.

We see that the Semantic Anomalies generally achieves the best AUROC across all digits. The integration often achieves the best AUPR but struggles with certain held out digits. In particular, the integration performs significantly worse on the "4" and "9" digits. This is a reasonable confuser due to the similarities and variability of the two classes. These results indicate that the integration penalties are helpful for inducing the model to detect semantic anomalies.

Table 7: MNIST leave one out: AUPR (%)

| Method | 0 | 1 | 2 | 3 | 4 |
|---|---|---|---|---|---|
| EGBAD | 78 | 29 | 67 | 52 | 46 |
| SA: Rotation | 89.24 | 83.57 | 78.20 | 66.33 | 90.45 |
| Global & Local | $95.26 \pm 1.98$ | $89.67 \pm 4.33$ | $86.02 \pm 3.19$ | $93.41 \pm 1.88$ | $69.89 \pm 7.02$ |

| | 5 | 6 | 7 | 8 | 9 |
|---|---|---|---|---|---|
| EGBAD | 43 | 57 | 35 | 54 | 35 |
| SA: Rotation | 83.38 | 75.57 | 95.19 | 68.84 | 84.88 |
| Global & Local | $87.76 \pm 2.1$ | $91.7 \pm 2.84$ | $83.43 \pm 3.72$ | $87.5 \pm 2.91$ | $72.9 \pm 7.23$ |

## C TRAINING AND ARCHITECTURES

### C.1 SPIRALS

The bijective layer is composed of five blocks where each block contains an affine-coupling layer (Dinh et al., 2017) and an invertible fully connected layer. The data distribution is bimodal over both latent dimensions with means at $\pm 1$ and standard deviations of $0.4$. The separable network is composed of quadratic functions. We train the model as discussed, using the standard cross-entropy

Table 8: MNIST leave one out: AUROC (%)

| Method | 0 | 1 | 2 | 3 | 4 |
|---|---|---|---|---|---|
| EGBAD | 78 | 29 | 67 | 52 | 45 |
| GANomaloy | 88 | 65 | 95 | 79 | 80 |
| SA: Rotation | 98.85 | 98.17 | 94.60 | 94.22 | 98.23 |
| Global & Local | $95.93 \pm 1.91$ | $89.32 \pm 4.76$ | $84.83 \pm 3.94$ | $94.25 \pm 1.25$ | $74.49 \pm 6.98$ |

| | 5 | 6 | 7 | 8 | 9 |
|---|---|---|---|---|---|
| EGBAD | 43 | 57 | 39 | 55 | 36 |
| GANomaloy | 85 | 85 | 68 | 85 | 55 |
| SA: Rotation | 97.21 | 92.52 | 99.16 | 92.69 | 97.02 |
| Global & Local | $89.47 \pm 2.32$ | $93.11 \pm 2.98$ | $83.45 \pm 4.01$ | $88.42 \pm 2.97$ | $75.66 \pm 5.65$ |

loss for each observed point, the negative log-likelihood over the input space, and the global cross-entropy integration with respect to the contrastive prior. The learning rate is set to 0.001 with standard weight decay of 1e-4 over 20 epochs using Adam (Kingma & Ba, 2015).

## C.2 FASHION MNIST

The bijective layers are composed of the Glow (Kingma & Dhariwal, 2018) architecture with two levels composed of 32 steps and 512 channels. We utilize an open-source Glow implementation[1] wrapped in Pyro's (Bingham et al., 2018) bijective operators. The data distribution is bimodal over all latent dimensions with means at $\pm 1$ and standard deviations of $0.5$. The noise constrastive distribution is a standard Gaussian. The separable network is composed of hinge functions (see Sec. 4.3). We utilize a batch size of 256, a learning rate of 1.5e-4 over the bijective layers, and a learning rate of 1.0e-3 over the separable layers using Adam (Kingma & Ba, 2015). Both learning rates are exponentially decayed at a rate of 0.75 per epoch. Standard weight decay is applied with a weight of 1e-4. The network is trained for 50 epochs. The adversarial attacks are performed against the model at the epoch with the lowest validation cross-entropy loss.

## D APPROXIMATE EXPECTED CROSS-ENTROPY OVER A DOMAIN

We desire the expected cross-entropy, $\text{CE}(\hat{y}(\mathbf{x}), y_c)$, over a domain $\mathcal{D}_c$ where $\mathbf{y}_c$ is the desired probability vector over classes, $\hat{y}(\mathbf{x})$ is the model's prediction of $y \in \mathbb{R}^K$ from $\mathbf{x} \in \mathbb{R}^M$ and is composed of a bijective function, $\mathbf{h}$, a separable function, $f_k(\mathbf{x}) = \sum_m f_{k,m}(x_m)$, and the soft-max function, $\sigma$, such that $\hat{\mathbf{y}}(\mathbf{x}) = \sigma(\mathbf{f}(\mathbf{h}(\mathbf{x})))$. If we are attempting to regularize the model to produce a single class label over $\mathcal{D}_c$, then $\mathbf{y}$ will correspond to a one-hot vector. In general, however, $\mathbf{y}_c$ may be a dense vector with elements $y_k$. The expected cross-entropy is expressed as

$$\mathbb{E}_{\mathcal{D}_c}\left[\text{CE}(\hat{y}(\mathbf{x}), y_c)\right] = -\sum_{k=1}^{K} y_k \, \mathbb{E}_{\mathcal{D}_c}\left[\log(\hat{y}_k(\mathbf{x}))\right] = -\sum_{k=1}^{K} y_k \int_{\mathcal{D}_c} \log(\hat{y}_k(\mathbf{x})) \, p_{\mathcal{D}_c}(\mathbf{x}|\mathbf{y})d\mathbf{x} \quad \text{(D1)}$$

where $p_{\mathcal{D}_c}(\mathbf{x}|\mathbf{y})$ is the probability density function over $\mathcal{D}_c$ conditioned on $\mathbf{y}$. We can take advantage of the bijective transform to convert this expression into an expectation over the latent space, $\mathbf{z}$, and latent domain, $\mathcal{Z}_c$, which allows us to write

$$\sum_{k=1}^{K} y_k \, \mathbb{E}_{\mathcal{D}_c}\left[\log(\hat{y}_k(\mathbf{x}))\right] = \sum_{k=1}^{K} y_k \, \mathbb{E}_{\mathcal{Z}_c}\left[\log(\sigma(\mathbf{f}(\mathbf{z}))\right] = \sum_{k=1}^{K} y_k \int_{\mathcal{Z}_c} \log(\sigma_k(\mathbf{f}(\mathbf{z}))) \, p_{\mathcal{Z}_c}(\mathbf{z}|\mathbf{y})d\mathbf{z}$$
$$\text{(D2)}$$

where $\sigma_k$ is the $k$th element after the soft-max normalization and $p_{\mathcal{Z}_c}(\mathbf{z}|\mathbf{y})$ is the independent probability density function in the latent space, e.g., $p_{\mathcal{Z}_c}(\mathbf{z}|\mathbf{y}) = \prod_m p_m(z_m)$. We can then expand

---

[1]https://github.com/chrischute/glow

the soft-max operator within the expectation to get

$$\sum_{k=1}^{K} y_k \, \mathbb{E}_{\mathcal{Z}_c} \left[ \log(\sigma(\mathbf{f}(\mathbf{z}))) \right] = - \mathbb{E}_{\mathcal{Z}_c} \left[ \log \left( \sum_{j=1}^{K} \exp(f_j(\mathbf{z})) \right) \right] + \sum_{k=1}^{K} y_k \, \mathbb{E}_{\mathcal{Z}_c} \left[ f_k(\mathbf{z}) \right] \qquad \text{(D3)}$$

The combination of the separable function and independent densities allows for an exact simplification of the second term into

$$\sum_{k=1}^{K} y_k \, \mathbb{E}_{\mathcal{Z}_c} \left[ f_k(\mathbf{z}) \right] = \sum_{k=1}^{K} y_k \sum_{m=1}^{M} \int_{\mathcal{Z}_m} f_{k,m}(z_m) p_m(z_m) dz_m \qquad \text{(D4)}$$

$$\approx \frac{1}{G} \sum_{k=1}^{K} y_k \sum_{m=1}^{M} \sum_{g=1}^{G} f_{k,m}(\tilde{z}_{m,g})$$

where we have overloaded $\mathcal{Z}_m$ to correspond to the domain of the $m$th latent dimension and have approximated the integral via Monte Carlo integration with $\tilde{z}_{m,g} \tilde{p}_m(z_m)$ and $G$ draws. The first term on the right-hand side of Eq. D3 does not enjoy the same exact simplification. However, it is possible to bound the term via Jensen's Inequality by moving the logarithm into the sum over $j$ or outside the expectation operator. We consider the latter case where

$$\mathbb{E}_{\mathcal{Z}_c} \left[ \log \left( \sum_{j=1}^{K} \exp(f_j(\mathbf{z})) \right) \right] \leq \log \left( \mathbb{E}_{\mathcal{Z}_c} \left[ \sum_{j=1}^{K} \exp(f_j(\mathbf{z})) \right] \right). \qquad \text{(D5)}$$

Then, we expand the separable function, exchange the order of the sum over dimensions and the exponent, take advantage of the multiplicative integral simplification (Eq. A5), approximate via Monte Carlo integration, and utilize the $\log \sum \exp$ operator for stability:

$$\log \left( \mathbb{E}_{\mathcal{Z}_c} \left[ \sum_{j=1}^{K} \exp(f_j(\mathbf{z})) \right] \right) = \log \left( \mathbb{E}_{\mathcal{Z}_c} \left[ \sum_{j=1}^{K} \exp(\sum_{m=1}^{M} f_{j,m}(z_m)) \right] \right) \qquad \text{(D6)}$$

$$= \log \left( \sum_{j=1}^{K} \int_{\mathcal{Z}_c} d\mathbf{z} \prod_{m=1}^{M} \exp(f_{j,m}(z_m)) \prod_{n=1}^{M} p_n(z_n) \right)$$

$$= \log \left( \sum_{j=1}^{K} \prod_{m=1}^{M} \int_{\mathcal{Z}_m} \exp(f_{j,m}(z_m)) p_m(z_m) dz_m \right)$$

$$\approx \log \left( \sum_{j=1}^{K} \prod_{m=1}^{M} \sum_{g=1}^{G} \exp(f_{j,m}(\tilde{z}_{m,g})) \right) - M \log(G)$$

$$= \log \sum_{j=1}^{K} \exp \sum_{m=1}^{M} \log \sum_{g=1}^{G} \exp(f_{j,m}(\tilde{z}_{m,g})) - M \log(G).$$

Finally, we can substitute Eq. D4 and Eq. D6 into Eq. D1 to get

$$\mathbb{E}_{\mathcal{D}_c} \left[ \text{CE}(\hat{y}(\mathbf{x}), y_c) \right] \leq - \sum_{k=1}^{K} y_k \sum_{m=1}^{M} \int_{\mathcal{Z}_m} f_{k,m}(z_m) p_m(z_m) dz_m \qquad \text{(D7)}$$

$$+ \log \left( \sum_{j=1}^{K} \prod_{m=1}^{M} \int_{\mathcal{Z}_m} \exp(f_{j,m}(z_m)) p_m(z_m) dz_m \right)$$

$$\approx - \frac{1}{G} \sum_{k=1}^{K} y_k \sum_{m=1}^{M} \sum_{g=1}^{G} f_{k,m}(\tilde{z}_{m,g})$$

$$+ \log \sum_{j=1}^{K} \exp \sum_{m=1}^{M} \log \sum_{g=1}^{G} \exp(f_{j,m}(\tilde{z}_{m,g})) - M \log(G)$$

In the event that $y_c$ is a one-hot vector (at the $c$th element) this simplifies to

$$\mathbb{E}_{\mathcal{D}_c}\left[\text{CE}(\hat{y}(\mathbf{x}), y_c)\right] \leq - \sum_{m=1}^{M} \int_{\mathcal{Z}_m} f_{c,m}(z_m) p_m(z_m|c) dz_m \tag{D8}$$

$$+ \log\left(\sum_{j=1}^{K} \prod_{m=1}^{M} \int_{\mathcal{Z}_m} \exp(f_{j,m}(z_m)) p_m(z_m|c) dz_m\right)$$

where we reintroduced the condition over the class within the probability densities $p_m(z_m|c)$ for emphasis.

