# OpenReview forum: "Practical Integration via Separable Bijective Networks"
_ICLR.cc/2022/Conference — ICLR 2022 Poster_

### Official Review · Reviewer_PTfm · 2021-10-27

**Correctness:** 3
**Technical Novelty And Significance:** 3
**Empirical Novelty And Significance:** 3
**Recommendation:** 6
**Confidence:** 4

**Main Review:**

The paper is clearly written and well-motivated. I think that integration over a continuous region -- instead of the more heuristic approach of random perturbation or augmentation -- is a novel and more principled strategy that enhances the robustness of models. For the purpose of tractable integration, using neural networks to parametrize separable functions is a natural approach.

My main concern, however, is the scope of application of this work. Based on Section 1-6 of the paper, my impression is that the authors aim to keep the method general-purpose. Indeed, when reviewing relevant work on OOD in Section 6, the authors mentioned that "These methods are constructed specifically for OOD detection whereas our method could be applied to a variety of problems based on the continuous regularizers chosen." With this impression in mind, I am slightly disappointed to find that all experiments in Section 7 are related to OOD detection. What is more, the proposed method is (slightly) worse than contemporary OOD detection baslines. For this reason, I find it difficult to assess the significance of the contribution of this work purely based on its performance on OOD detection. To showcase the claimed versatility of the proposed method, I suggest the authors include tasks other than OOD detection, or perhaps explain why only OOD detection is pertinent to the proposed method.

**Summary Of The Paper:**

In their submission, the authors proposed a computationally tractable way to perform integration over a high-dimensional region that the data samples reside. This integration technique is applied to OOD detection tasks.






**Summary Of The Review:**

To summarize, I think the current paper put forward an interesting idea (perform computationally tractable integration using separable functions parametrized by neural nets). However, it is unclear (at least based on the current presentation) whether the idea finds practical application in machine learning. The experiment section of this paper focuses on OOD detection, in which case the proposed method performs OK-ish but fails to compete against contemporary baseline methods. Thanks to its generality, the proposed integral method may be useful for other ML applications, which I encourage the authors to demonstrate.

---

> ### Author Response · Authors · 2021-11-19
> **Response to Reviewer PTfm**
>
> Thank you for your review and feedback. We are pleased that you agree that supervision over continuous regions is a better strategy than ad hoc perturbation or augmentation. Based on your comments, it seems that your main concern is that focused on OOD detection when the proposed method is more generally applicable. We found your comments very appropriate and have addressed them in the general response to emphasize it to other reviewers and the community in general.

---

> > ### Comment · Reviewer_PTfm · 2021-11-26
> > **Response to authors**
> >
> > Thanks for your response. I appreciate the new semi-supervised experiment over tabular MNIST. Although the results seem preliminary, they indeed showcase the utility of the proposed method beyond OOD. I have raised my score accordingly.
> >
> > In the meanwhile, I suggest the authors fully work out this semi-supervised tabular MNIST experiment and include it in the paper with baselines and experimental details. It seems this experiment is only described in the authors' response and not included in the up-to-date version of the paper.

---

### Official Review · Reviewer_mYvf · 2021-10-28

**Correctness:** 3
**Technical Novelty And Significance:** 1
**Empirical Novelty And Significance:** 1
**Recommendation:** 1
**Confidence:** 3

**Main Review:**

The writing renders the connection between (\*) and (\*\*) very obscure. After all, commonly-encountered $\Omega \circ \hat y$ in ML/DL cannot be written as the composition of a separable function and bijective function. In other words, it is difficult to discern the relevance of (**) when the goal is to estimate (*).

Reading Appendix D helped clarify this confusion. The paper operates under the assumption that the model prediction $\hat y$ can be written as the composition of a bijective $h$, a separable $f$ and the soft-max function $\sigma$, i.e., (***) $\hat y(x) = \sigma(f(h(x)))$. This is a big assumption and should not be relegated to the Appendix. It needs to be front and center in the main text. Even under this assumption the expected cross-entropy in (D1) still cannot be written as (\*). The best we can do with this (\***) assumption is to derive an upper bound on (\*). This is really quite underwhelming. Give me MC error over a potentially very un-tight bound  any day.

Besides the exposition, I hold reservations about the proposed methodology. It is not clear that it’s a worthwhile to restrict a neural network classifier to be of the form (***) only to arrive at the bound in Equation D8 (which is same as Equation 13 in the main text).

**Summary Of The Paper:**

For input-output pair $(x,y)$, the paper undertakes the task of estimating (*) $E_{x \sim p(x)} \Omega(\hat y(x))$. When $x$ is high dimensional, integration is difficult. Standard ML/DL applies MC to estimate this integral based on an iid sample $(x_i,y_i), i=1,\ldots,n$. This paper suggests there is a better way.

The main idea is to recognize that if $f$ is a separable function and $h$ is a bijective function, then (**) $E_{x \sim p(x)} f(h(x))$ is easy to evaluate.

**Summary Of The Review:**

Although I like the idea that we should pay more attention to integration error, I don't see how we can get around it in ML/DL. The proposed approach of assuming (\***) is certainly not the way. Furthermore, the function $\Omega$ in (\*) that would be encountered in ML/DL cannot be accounted for in the proposed framework unless some type of bound is in D8 is employed. Replacing MC error with an upper bound on a nice loss function as in D8 simply does not seem like a good trade to me.

---

> ### Author Response · Authors · 2021-11-19
> **Response to Reviewer mYvF**
>
> tl;dr: The concerns raised are largely inconsistent with common and successful methodology in ML and ignore the strength of our experimental results. Optimizing a bound in lieu of MC integration is a hallmark of Variational Inference and is well-established practice within the ML community. The integral of many losses can be expressed without bounds/approximations using separable functions. Normalizing flow-type bijectors provide rich feature extractors that represent one’s distribution of complicated inputs with simply-distributed, independent features. Thus, separable functions over the learned independent features shall be expressive (especially when compared to the previous standard approach of constructing separable functions directly in the input features). We have demonstrated that separable functions provide similar performance to non-separable functions when coupled with a sufficiently-complex bijector which demonstrates that they can be a reasonable choice.
>
> Thank you for your comments and especially for reviewing the appendices. We enumerate and address issues below. We believe that most of the concerns raised may be addressed by considering analogues standard ML approaches. We hope that considering the established methodology for lower bounds and functional approximation clarifies any gaps and allows you to reconsider your score.
>
> You find several of the key points difficult to connect within the main text, most especially, the composition of the softmax, separable, and bijective components, $\sigma(f(h(x)))$, and the connection between expectations relative to the input and latent spaces.
>  - We have modified the document to render the connection between the expectations more explicitly. The formulation of $y$ as a composition of softmax, separable, and bijective functions is included in the paper (both in words and as an inline equation) but we have similarly modified the document to add greater emphasis.
>
> You do not think the tradeoff between MC error and a bound is worthwhile.
>  - The tradeoff and use of lowerbounds versus MC integration has been well studied and employed in variational inference. Most notably, the use of a bound instead of an MC integrator is an essential piece of the celebrated variational autoencoder. This class of models optimizes with respect to the evidence **lower bound** (ELBO) rather than performing a high-dimensional integral to marginalize out the latent code. Our use of optimizing a bound is consistent with this practice and is empirically validated for the examples we explored (optimizing the bound results in improvements in OOD detection *in all cases*). Further, *all* variational inference utilizes the ELBO as the optimization goal instead of MC integrators.
>
>
> You do not believe that many network outputs or desirable regularizers can be modeled as a composition of separable and bijective functions.
>
>   i.) We intentionally chose a common problem setting/formulation that is not separable (since the outputs must sum to one) as a stress-case of our method and to illustrate a possible way to circumvent the mismatch. Our overall goal is to operate with an approximation to functions that are not separable in the input space, via a separable function in a latent space. This approximation, like any other parametric/finite sample approach, incurs a bias; however, it also enables us to integrate over the domain explicitly for regularization and other purposes. Our approach is actually much less biased than other separable estimators, which typically fit a separable function directly on the input features, yet still obtain useful models (e.g. see additive models such as [1]). We expound below.
>
>   ii.) It is a common assumption in tractable-likelihood models that the data distribution can be represented as a product of independent distributions within the latent space [2,3,4]. Sufficiently powerful bijectors disentangle the complex interactions between input features to enable this separable representation. Generalizing this to a discriminative (or other) process from that independent latent space is entirely self-consistent.
>
>   iii.) The use of a composition of a separable and bijective function achieves comparable performance with models that consider the composition of a generic function and a bijective network (Sec. 7.2). This indicates that our formulation provides a good approximation for the true (unknown) function between $\mathbf{x}$ and $\mathbf{y}$. Please keep in mind that ML models are approximations (e.g., the ground-truth model is never perfectly parameterized as a finite network) and can only be validated empirically (in most cases).

---

> > ### Author Response · Authors · 2021-11-19
> > **Reference overflow**
> >
> > [1] Ravikumar, Pradeep, et al. "Sparse additive models." Journal of the Royal Statistical Society: Series B (Statistical Methodology) 71.5 (2009): 1009-1030.
> >
> > [2] Laurent Dinh, Jascha Sohl-Dickstein, and Samy Bengio. Density estimation using real NVP. In 5th
> > International Conference on Learning Representations, ICLR 2017, Toulon, France, April 24-26,
> > 2017, Conference Track Proceedings. OpenReview.net, 2017.
> >
> > [3] Durk P Kingma and Prafulla Dhariwal. Glow: Generative flow with invertible 1x1 convolutions. In Advances in Neural Information Processing Systems, pp. 10236–10245, 2018.
> >
> > [4] Will Grathwohl, et. al. FFJORD: Free-form continuous dynamics for scalable reversible generative models. In International Conference on Learning Representations, 2019.

---

> > ### Comment · Reviewer_mYvf · 2021-11-21
> > **thank you for your comments**
> >
> > *The concerns raised are largely inconsistent with common and successful methodology in ML and ignore the strength of our experimental results. Optimizing a bound in lieu of MC integration is a hallmark of Variational Inference and is well-established practice within the ML community.*
> >
> > I respectfully disagree, in particular with your example. Variational inference does not optimize a bound in lieu of MC integration. After all there is an expectation $E_q$ that is taken over the variational distribution $q$ which is always approximated using MC integration. Maximizing the ELBO is equivalent to minimizing the KL divergence between the variational distribution and the (unnormalized) target density. It's not as if the ELBO is some bound on KL divergence. Rather it's a lower bound on the model evidence.
> >
> > *The integral of many losses can be expressed without bounds/approximations using separable functions.*
> >
> > I'm sorry if I'm mistaken, but your Equation (14) contradicts this?
> >
> > *You do not think the tradeoff between MC error and a bound is worthwhile...The tradeoff and use of lowerbounds versus MC integration has been well studied and employed in variational inference. Most notably, the use of a bound instead of an MC integrator is an essential piece of the celebrated variational autoencoder. This class of models optimizes with respect to the evidence lower bound (ELBO) rather than performing a high-dimensional integral to marginalize out the latent code. Our use of optimizing a bound is consistent with this practice and is empirically validated for the examples we explored (optimizing the bound results in improvements in OOD detection in all cases). Further, all variational inference utilizes the ELBO as the optimization goal instead of MC integrators.*
> >
> > I'm afraid I have to disagree again on your characterization of variational inference. The analogy does not feel relevant to me. What the ELBO is to the partition function is *not at all* what the RHS of Equation (14) is to Equation (1).
> >
> > *You do not believe that many network outputs or desirable regularizers can be modeled as a composition of separable and bijective functions.*
> >
> > That's not exactly what I meant to convey. Let me try to explain better. The very first equation of your paper, Equation (1), planted the seed in my mind that you wish to address the estimation of (1) for commonly-used losses $\Omega$ and commonly-used classifier $\hat y$. I think it's accurate to say most classifiers in use are *not* compositions of separable and bijective functions. (You say this yourself in the paper many times.) My main concern is that you do not advance knowledge in the estimation of (1) for the very simple fact that most commonly-used classifiers are not compositions of separable and bijective functions.
> >
> > *The use of a composition of a separable and bijective function achieves comparable performance with models that consider the composition of a generic function and a bijective network (Sec. 7.2). *
> >
> > Alright I am certainly willing to entertain that. But that is not the point of your paper. Your work proposes a better way of estimating Equation 1 (?). Whether classifiers should always be separable $\circ$ bijective is a different matter from estimating (1) when $\hat y$ (or $\Omega \circ \hat y$) is not this special composition.
> >
> > I have not yet read the revision in detail. From the response here, I do hope the authors did not rely too much on the analogies with VI in their revision...

---

> > > ### Author Response · Authors · 2021-11-22
> > > **Thank you for the quick response**
> > >
> > > Thank you for your quick response. Please find responses to points below:
> > >
> > > > Variational inference does not optimize a bound in lieu of MC integration.
> > >
> > > Variational inference for VAEs is exactly for this case. That is, in a VAE, where the $p(x)$ stems from the intractable $\int p(x|z) p(z) dz$. A naive approach would be to use MC integration and directly estimate as $\frac{1}{M} \sum_{i=1}^M p(x|z_j)$ with samples $z_j \sim p(z)$. Instead, a lower bound is optimized in lieu of MC integration. E.g. please see https://arxiv.org/pdf/1606.05908.pdf for more details.
> > >
> > > > I'm afraid I have to disagree again on your characterization of variational inference. The analogy does not feel relevant to me.
> > >
> > > We did not state that we were performing variational inference. We set forth the variational Inference analogy to address your concern, "Give me MC error over a potentially very un-tight bound any day." Our use of a bound for classification (just one application of our approach) yielded successful predictions, which would suggest that the bound is not overly "un-tight." Thus, perhaps one finds issues with a bound itself. We present variational inference as an example standard approach that successfully employs bounds.
> > >
> > > > Most commonly-used classifiers are not compositions of separable and bijective functions.
> > >
> > > This is not a pertinent point to our work. Our method is intended to provide a practical way to estimate the integral of $\Omega$ (for all applications, not just classification), which commonly-used estimators are unable to yield. Hence, this paper presents a family of architectures that make the evaluation of $\int \Omega$ possible without exponential cost. Moreover, most real-world dependencies are not the result of finite parameterized commonly-used classifiers. There are always biases to finite-sample/parametric approaches. However, as explained in our original response, the composition of normalizing flow bijectors and separable functions are expressive (as further evidenced by our results).
> > >
> > > > The integral of many losses can be expressed without bounds/approximations using separable functions.
> > > > I'm sorry if I'm mistaken, but your Equation (14) contradicts this?
> > >
> > > Equation 14 is unique to one loss/application (cross-entropy/classification). As mentioned by reviewer PTfm and discussed in our general response, the method we present has greater reach than just the CE loss and classifiers. Many other applications will be more straightforward. In fact, we focus on this problem setting precisely because of this difficulty.

---

### Official Review · Reviewer_6qXG · 2021-11-02

**Correctness:** 4
**Technical Novelty And Significance:** 3
**Empirical Novelty And Significance:** 3
**Recommendation:** 6
**Confidence:** 4

**Main Review:**

The method uses a bijective flow network to map dependent input features to a set of independent latent variables. It then applies a separable function on the latent variables to get a loss function (or an approximation). The combination results in an integral that decomposes into separate 1-d integrals across each dimension. This makes it possible to calculate the gradient of the loss function efficiently across the latent domain.

The empirical results are mixed. The use of regularizers based on the decomposable integrals is shown to be beneficial for OOD detection, but the results cannot always match the baseline (likely due to the constraints on the network structure). The results of experiments with adversarial data (in the appendix) look less interesting and might suggest that the method is not effective against targeted adversarial attacks. The description of the experiments is minimal in the main body of the paper and in parts hard to follow.

Note: I was a reviewer to an earlier version of this work. Compared to the previous version, the writing is improved in the parts and the experiments are expanded and rearranged.

Minor notes:
- It’s more accurate to present the sample complexity of the separable integral as $O(GM)$ when comparing to $O(G^M)$.
- Theorem names do not match between the main body and the appendix.
- The notation in section 5.3 does not match the discussion on separable functions in section 3.2.1. In particular, it is better not to use $x_m$ as the argument (maybe $z_m$ or $v_m$). It’s also better to make the parameters of the network explicit in the formulas (11): e.g. $f_{k,m}(z_m; u_{k,m}\nu_{k,m}) = \dots$
- Add names to the first two columns of Table 2.


**Summary Of The Paper:**

This paper introduces a hybrid model architecture that makes it possible to integrate a separable loss function across a region of input space. Such integrals can be used as regularizers for robustness near the observed data points (local consistency), and out-of-distribution (OOD) detection in neighborhoods away from the observations. The paper uses these regularizers to train models that are less vulnerable to adversarial attacks and out of distribution mishaps.

**Summary Of The Review:**

Originality and significance: The combination of flow networks with separable functions to form tractable integrals is a novel idea (to the best of my knowledge), with potential use in other ML domains. The experiments on OOD detection tasks show parity with SOTA baseline in some but not all the studied cases.

Quality and clarity: The paper is hard to follow in parts, mostly due to the fact that it is not self-contained. The reader is expected to know the background on bijective networks. The experimental section is lacking details in the main body of the paper.

---

> ### Author Response · Authors · 2021-11-19
> **Response to Reviewer 6qXG**
>
> Thank you for your comments and suggestions. We are very much appreciative of your continued feedback. Per your request, we have modified Sec. 3.1 to increase the discussion about bijective networks. We are also willing to introduce an additional section in the appendix to provide further elaborations if this would be helpful.
> Due to space constraints, we include all experimental details in the appendix (App. B).
> We have also included the specific corrections/clarifications that you mentioned in your review.
> Please see the general response for a more in-depth discussion.

---

> > ### Comment · Reviewer_6qXG · 2021-11-27
> > **Comment**
> >
> > The added context on bijective networks has made the paper more readable. I still think the complexity should be described as $O(GM)$ and not $O(G)$.

---

> > > ### Author Response · Authors · 2021-11-29
> > > **O(GM)**
> > >
> > > We intended for this correction when changing the assessment from the number of network evaluations ($\mathcal{O}(G)$) to the overall complexity ($\mathcal{O}(GM)$) in the beginning of Section 4. We will correct this transcription error and note the overall complexity at $\mathcal{O}(G)$ in future versions.

---

### Official Review · Reviewer_MsPk · 2021-11-03

**Correctness:** 3
**Technical Novelty And Significance:** 3
**Empirical Novelty And Significance:** 3
**Recommendation:** 8
**Confidence:** 2

**Main Review:**

The paper is clearly written and relatively easy to follow.  The method combines known methodologies in a relatively straightforward but novel manner to achieve the desired goal, which is well-motivated.  The exposition is clear and the experimental section covers a good number of compelling applications for the method.

One potential issue with this paper is that the method described simply combines known approaches.  Overall I don't consider this a particularly serious flaw but it is something to consider.

**Summary Of The Paper:**

The paper proposes using separable bijective networks - that is, a two-stage network where the first stage is bijective (data-flow either way) and the second stage is separable (multiplicative or additive) - to make it practical to include integrals (on the input) as performance goals.

**Summary Of The Review:**

Good paper combining known methods to solve a well-motivated problem with a number of compelling applications.

---

> ### Author Response · Authors · 2021-11-19
> **Response to Reviewer MsPK**
>
> Thank you very much for your comments and are pleased that you found the method and motivation accessible and intuitive. We hope that others appreciate the novelty in the composition of separable and bijective functions to enable learning over continuous regions and the construction of novel regularizers.

---

### Author Response · Authors · 2021-11-19
**General Comments**

tl;dr: We focus on OOD detection because it is a familiar model and a good stress test for our method. We perform a semi-supervised experiment over tabular data and achieve similar performance to the state of the art as demonstration of greater applicability.

We would like to thank all of the reviewers for their efforts and feedback. We have uploaded a new version of the document with the requested modifications (changes emphasized via blue text, removals included via strikethrough to ease comparison). We address specific questions and concerns in separate comments directed to each reviewer.

We opted to focus on OOD detection since this setting is accessible/familiar to the ML community and provides a stress-case due to our separable formulation and desire to supervise unobserved regions, i.e., the use of the softmax normalization makes it impossible to write the output as a separable network and OOD examples, by definition, do not exist in the training set. This forces us to provide an example on how to deal with cases where typical formulations would seem to preclude our method. We deal with this by minimizing an upper bound and show that this is effective in all cases (OOD detection is improved with integrated regularization). Similarly, this setting allows us to discuss how we can modify the latent distributions to include a contrastive prior and then showcase how to exploit that prior to regularize the model.

However, we do agree with reviewer PTfm that alternative problem settings would help to showcase the broader applicability with less strenuous requirements. We are particularly interested in applying local integrals to semi-/self-supervised problems where it is not clear how to naturally augment the data (e.g.,general tabular data). We attempted a quick test using pseudo-labeling [1] over a flattened (tabular) version of MNIST. This is a standard choice for semi-supervised methods outside of the image/text domain [2]. Since we can no longer take advantage of image-specific augmentations (because the data is a flattened vectors of real values) many standard methods become inaccessible [3]. We train using 10% of the labels and the same local integrals in the main paper (without the reject option or corresponding global integral). The bijector is a CNF [4] and the separable function is a hinge. We achieve a validation accuracy of *95.3%* and expect additional hyperparameter tuning would result in further improvements. To our knowledge, the current state of the art on this problem using semi-supervised methods is achieved by VIME [2] at *95.5%* with semi-supervised methods and *95.7%* with additional self-supervised methods. We hope this simple test illustrates the greater reach of the method beyond just OOD.

[1] Dong-Hyun Lee. Pseudo-label: The simple and efficient semi-supervised learning method for
deep neural networks. In ICML Workshop on Challenges in Representation Learning, 2013.

[2] Jinsung Yoon, et. al. VIME: Extending the Success of Self- and Semi-supervised Learning to Tabular Domain. In Advances in Neural Information Processing Systems, pp. 11033-11043, 2020.

[3] David Berthelot, et. al. Mixmatch: A holistic approach to semisupervised learning. In Advances in Neural Information Processing Systems, pp. 5050–5060, 2019.

[4] Will Grathwohl, et. al. FFJORD: Free-form continuous dynamics for scalable reversible generative models. In International Conference on Learning Representations, 2019.

---

### Decision · Program_Chairs · 2022-01-20

**Decision:**

Accept (Poster)

**Comment:**

This paper suggests the use of networks for supervised learning which are composed of a bijective network (e.g. a flow) followed by a separable function. This allows easy integration over the input space, which can be used to formulate novel regularizers (examples given are for local consistency, and for out-of-distribution detection).

The approach is pretty novel, and it's an interesting paper. The reviewers were very divided, however; one reviewer giving it a 1, and another an 8, with the other two reviewers arguing weakly to accept. The "1" took issues with the general formulation, feeling that the necessity to optimize bounds on the true objectives in cases such as softmax regression greatly limits the viability of the work. Personally I disagree with that reviewer's characterizations of the novelty and significance of this work, and I think the OOD detection / classification experimental setting is sufficient to make their point that their approach can be applied in such settings.

In the end I (AC) would agree with the "weak accept" / 6, given the draft of the paper at this time. While I think a few things could be presented more clearly, and I think the empirical evaluation could be more robust (e.g. exploring what goes wrong when the integrated objective functions are included on CIFAR and SVHN), and it would be nice to explore additional applications, I think this is a creative paper which it would be nice to include at ICLR.